# A Fading Affect Bias First: Specific Healthy Coping with Partner-Esteem for Romantic Relationship and Non-Relationship Events

**DOI:** 10.3390/ijerph181910121

**Published:** 2021-09-26

**Authors:** Jeffrey Alan Gibbons, Spencer Dunlap, Kyle Horowitz, Kalli Wilson

**Affiliations:** Department of Psychology, Christopher Newport University, 1 Avenue of the Arts, Newport News, VA 23606, USA; spencer.dunlap.14@cnu.edu (S.D.); kyle.horowtiz.13@cnu.edu (K.H.); kalli.wilson.11@cnu.edu (K.W.)

**Keywords:** fading affect, coping, romantic relationships, partner-esteem, rehearsal

## Abstract

The Fading Affect Bias (FAB) is the faster fading of unpleasant affect than pleasant affect. Research suggests that the FAB is an indicator of general healthy coping, but it has not shown consistent specific healthy coping via differential relations of the FAB to individual differences across event types. Although previous research did not find specific healthy coping for the FAB across romantic relationship events, these researchers did not include non-relationship control events. Therefore, we examined the relation of the FAB to various relationship variables across romantic relationship events and non-relationship control events. We found general healthy coping in the form of robust FAB effects across both event types and expected relations between relationship variables and the FAB. We also found three significant three-way interactions with the FAB showing specific healthy coping for partner-esteem, which is novel for the FAB. Rehearsal ratings mediated all the three-way interactions.

## 1. Introduction

Whereas the beginnings of a new romantic relationship can be so joyous that they produce feelings of euphoria [1], breaks ups can produce such strong unpleasant emotions that they parallel, and even rival, feelings produced by the death of a loved one [2]. Regardless of individuals’ past romantic relationship outcomes, people spend a great deal of time, energy, and effort moving past their previous relationships, maintaining their current ones, and seeking new relationships, as these interactions are demanding and emotionally intense [3], especially in the case of betrayals (Finkel et al., 2002 [4]; Luchies et al., 2013 [5]). Based on autobiographical memory research showing that unpleasant affect fades faster than pleasant affect [6,7], which is referred to as the Fading Affect Bias FAB: [8], romantic relationship events should show strong FAB effects. In fact, Zengel and his colleagues [3] examined and found the FAB across sexual and non-sexual relationship events, which showed general healthy coping. However, the researchers may not have shown specific healthy coping, in the form of significant but different FAB effects across events, because they did not examine non-relationship control events. Therefore, the current study tested for specific healthy coping in the context of romantic relationships by investigating the relation of the FAB to relationship variables across romantic relationship and non-relationship events.

### 1.1. Fading Affect Bias

The seminal autobiographical memory research examining the emotions tied to personal events demonstrated a positivity bias in memory using diary procedures. Specifically, more pleasant events were recalled than unpleasant events [9,10,11,12], and their affect faded faster for unpleasant events than for pleasant events [6,13]. Other researchers also used diary procedures and found that affect initially faded within the first 12 to 24 h [14] and it faded more over time for unpleasant than for pleasant events [7]. This differential fading of emotions was deemed the Fading Affect Bias (FAB) by Walker and colleagues [8], and it increased with retention intervals longer than 3 months [7,15].

The FAB is positively related to social rehearsals [16,17], and listener type moderates this effect, such that this relation is stronger for interactive than for non-responsive listeners [18]. Moreover, the FAB is moderated by several emotions, which typically disrupt the FAB, such as dysphoria [8,19], dispositional mood [20], trait anxiety [21], state anxiety [19,22], and stress [19,22]. These results support the mobilization-minimization hypothesis, which suggests that biological, cognitive, and emotional resources are galvanized to reduce the harmful effects of unpleasant circumstances [23]. The FAB phenomenon persists across a variety of cultures, and it may help people seek pleasant experiences, avoid unpleasant ones [24], and put the unpleasant experiences in perspective [25], which enhances self-perceptions and regulates emotions [15,26]. In other words, the FAB seems to be a general healthy coping outcome that helps people feel good about themselves.

Whereas general healthy coping is shown via robust FAB effects across events as well as positive and negative relations of the FAB to healthy and unhealthy outcomes, respectively, specific healthy coping is much more complex and difficult to display. For specific healthy coping, the FAB must be positively related to some healthy outcome or negatively related to some unhealthy outcome, and, across event types, these relations must be reliable and differ in strength as suggested by Gibbons and colleagues [27]. In contrast to the vast FAB research consistently showing general healthy coping e.g., [19,25], this literature has not shown specific healthy coping across event types and individual differences.

Whereas research has not yet shown that the FAB is a healthy coping outcome at the specific level of analysis, we remained hopeful that we could provide such evidence in the context of romantic relationships. In fact, we hoped that Zengel and colleagues [3] would provide evidence of specific healthy coping when they examined the relation of the FAB to various moderators across sexual and non-sexual relationship events. Although the researchers found general healthy coping in the form of significant FAB effects for both events as well as various moderator effects for the FAB that also showed general healthy coping, the researchers did not show specific healthy coping. In other words, the FAB was not positively related to a healthy outcome or negatively related to an unhealthy outcome more strongly for one event (e.g., sexual) than for the other event (e.g., non-sexual). One possible reason for this outcome is that Zengel and colleagues did not test non-relationship control events. Therefore, we planned to evaluate the relation of the FAB to important relationship variables across relationship and non-relationship events. We first looked to the literature on romantic relationships to determine important relationship variables that might predict the healthy outcome that is the FAB.

### 1.2. Romantic Relationships Are Related to Esteem, Satisfaction, Confidence, and Attachment

The literature on relationships showed that several variables were positively related to healthy outcome variables including self-esteem [28,29], partner-esteem [30], and relationship satisfaction [31,32]. Other relationship variables positively related to healthy outcome variables included relationship confidence [33] and secure attachment [34,35]. In contrast, the relationship literature showed that insecure attachment in the form of avoidant attachment was negatively related to healthy outcome variables [35,36,37], and ambivalent-anxious attachment was negatively related to healthy outcome variables [38] and positively related to unhealthy outcome variables [35].

## 2. The Current Study

Research has consistently demonstrated robust FAB effects, where unpleasant affect fades faster than pleasant affect [7,8]. The FAB is a general healthy outcome that encourages people to pursue pleasant experiences and buffer unpleasant ones [24,39]. The FAB is a general healthy outcome because it is positively related to healthy outcome variables, and it is negatively related to unhealthy outcome variables. However, the FAB has not indicated specific healthy coping because it has not been positively/negatively related to healthy/unhealthy variables, respectively, more strongly for one event (e.g., religious) than for another event (e.g., non-religious) across research examining various event types e.g., [25,27,40]. Although we hoped that Zengel and colleagues [3] would examine and show specific healthy coping for relationship events, they did not, which may have been due to their comparison of sexual events to non-sexual relationship control events rather than non-relationship control events. Consequently, we designed a study that examined the FAB across relationship and non-relationship events and we consulted the literature on romantic relationships to derive relevant relationship variables that could potentially produce specific healthy coping. The relationship variables included self-esteem, partner-esteem, relationship satisfaction, relationship confidence, as well as secure and insecure attachment.

We asked participants to provide ratings about their self-esteem, partner-esteem, relationship satisfaction, relationship confidence, and secure mother, father, and peer attachment. More importantly, we asked participants to provide brief event descriptions of 12 pleasant and unpleasant relationship and non-relationship events along with initial and current ratings of emotional affect as well rehearsal frequency ratings for those events. We controlled for neuroticism and the nominal participant variable because Gibbons and colleagues controlled for these variables as neuroticism is negatively related to the FAB and the multiple events provided by participants necessitate that participant is statistically controlled [27,41,42]. We expected to observe evidence of general healthy coping in the form of robust FAB effects for both relationship and non-relationship events as well as positive relations of the FAB to self-esteem, partner-esteem, relationship satisfaction, relationship confidence, as well as mother, father, and peer attachment. More importantly, we expected to see evidence of specific healthy coping, which would be the first instance of it in the FAB literature. For example, we expected to find that the FAB was positively related to partner-esteem and this relation would be stronger for romantic relationship events than for non-relationship events. We also expected rehearsal ratings to mediate any three-way interactions (as depicted in Figure 1) that were found in the current study (Gibbons et al., 2016 [41], Gibbons et al., 2015 [27]; Gibbons et al., 2013 [42]).

## 3. Method

### 3.1. Participants

The current study included only responses with no missing data, resulting in complete responses from 231 undergraduate students at a small southeastern liberal arts university. The students, ranging in age from 18 to 30 years old (*M* = 19.416, *SE* = 0.028), were primarily recruited from introductory psychology courses and were given course credit (when participation was a mandatory assignment in the class) or extra credit (when participation was not a mandatory assignment in the class) for their participation. Most of the sample was comprised of Caucasian (75.5%) women (77.9% women), who were predominantly Christian (77.5%) and heterosexual (94.4%). The study received approval from the Institutional Review Board at the university, which ensured that the procedure included a briefing, signed consent, and a debriefing as part of the American Psychological Association (APA) guidelines [43].

### 3.2. Materials and Measures

The questionnaires assessed general demographic information (e.g., age, race, sex, religion, and sexual orientation), neuroticism, relationship measures (e.g., self-esteem, partner-esteem, relationship satisfaction, relationship confidence, and attachment), and events (relationship and non-relationship). The event questionnaire asked for time of occurrence, a description, initial and test affect ratings, and rehearsal ratings.

**Neuroticism.** Although participants completed the entire brief version of the Big Five, called Mini Markers [44], we targeted the neuroticism because it has been negatively related to the FAB [27,41,42]. The subscale asks participants to rate the extent that they believe in the accuracy of self-descriptive adjectives. The eight items from the neuroticism component of the scale were evaluated, reversed scored when necessary (two items), and used to calculate an average score in the current study. High scores indicated high neuroticism. Cronbach’s alpha for the neuroticism scale was 0.706.

**Rosenberg Self-Esteem Scale (RSES).** The Rosenberg Self-Esteem Scale (RSES) measures self-worth with a combination of 10 questions pertaining to positive and negative feelings about one’s self [45]. Certain items were reverse scored with low scores indicating low levels of self-esteem. The scores from the items were averaged and Cronbach’s alpha for the self-esteem scale was 0.885.

**Rosenberg Partner-Esteem Scale (RPES)**. The Rosenberg Partner-Esteem Scale was used because of the interdependence of relationships, and the way one person’s emotion, cognition, or behavior can affect their partner’s emotion, cognition, or behavior [46]. Partner esteem was adapted from the Rosenberg Self-Esteem Scale (RSES), which uses 10 questions pertaining to positive and negative feelings about one’s self [45]. Certain items were reverse scored with low scores indicating low levels of self-esteem. The scores from the items were averaged and Cronbach’s alpha for the RPES scale was 0.830.

**Relationship Satisfaction Scale (RSS).** The Relationship Satisfaction Scale (RSS) uses 7 statements pertaining to relationship satisfaction [47]. Certain items were reverse scored with low scores indicating low levels of relationship satisfaction. The scores from the items were averaged and Cronbach’s alpha for the RSS scale was 0.890.

**Relationship Confidence Scale.** The relationship confidence scale (RCS) included 60 items with 25 of them being reversed scored [48]. All items were then averaged and Cronbach’s alpha for the RCS scale was 0.895.

**Inventory of Parent and Peer Attachment (IPPA).** The Inventory of Parent and Peer Attachment scale (IPPA) is a 75-question measure consisting of three sub-parts containing 25 items each: the Mother Attachment Scale (MAS), the Father Attachment Scale (FAS), and the Peer Attachment Scale (PAS). These scales each contain the same 25 items in the same order, and they measure attachment to one’s mother, father, and peers, respectively [34]. Certain items for each of the three 25-item scales are reverse scored for each of the scales with low scores indicating low levels of attachment. The average scores for mother, father, and peer attachment were calculated and Cronbach’s alpha for these measures were 0.946, 0.955, and 0.940, respectively.

**Fading affect and rehearsal for events.** The questionnaire prompted participants to describe 12 events: 3 pleasant relationship events, 3 unpleasant relationship events, 3 pleasant non-relationship events, and 3 unpleasant non-relationship events. The order of the four event types was counterbalanced in a Latin square. As each event type included three events, participants described and rated all three of the events for an event type and then they moved on to the next event type in the Latin square. Each event was rated for initial and current affect on a single-item scale. For initially pleasant events, fading affect was calculated by subtracting the current affect from the original affect. For initially unpleasant events, fading affect was calculated by subtracting the original affect from the current affect. These calculations ensured that all measures of fading affect were positive across pleasant and unpleasant events, such that a large fading affect score indicated a large amount of fading, and a small fading affect score indicated little fading. Each event was also rated for the frequency it was thought or talked about with a single-item scale.

We initially examined fading affect for the 2772 events provided by participants, but they did not provide affect ratings for some events (*N* = 22, 0.794%), which left 2750 events. Some unpleasant events were rated as pleasant and vice versa (*N* = 13, 0.469%), which left 2737 events. The initial scale also allowed for the possibility that events could be rated as initially neutral (*N* = 2, 0.072% of remaining events), leaving 2735 events. Some events increased in affective intensity (*N* = 180, 6.494%), which is referred to as flourishing [49], and other events switched their affective intensity to affect that was the opposite of the initial event affect (*N* = 290, 10.462%), which is referred to as changed affect [20]. Although these data have been removed in past studies, communications with Skowronski [50] via a review suggested that the FAB occurs regardless of the way that the data are analyzed. Using all the data enhances statistical power and, more importantly, it enhances generalizability. Therefore, we chose to analyze the data in all the different ways that affect can change or remain stable.

### 3.3. Procedure

Participants either signed up for the study through the online research management system SONA when completing the study for extra credit or they completed the study during their class time when completing the study for in-class participation points. At the start of the test, participants were given a consent form to read and then they were briefed by a researcher using a typed script. The researchers then informed the participants that they would be completing a series of questionnaires with specific instructions for completion above each one and then the researcher answered participants’ questions. Afterwards, participants signed the consent form.

After the briefing, the participants completed a questionnaire battery during a single sitting. The initial questionnaire consisted of general demographic information and various measures that assessed factors, such as relationship confidence, relationship satisfaction, self- and partner-esteem, and parent and peer attachment. In addition, the researchers collected information from participants regarding three events for each of four event types previously mentioned occurring in the past year. A relationship event was defined as anything that occurred in the context of the relationship between romantic partners, whereas a non-relationship event was defined as anything that occurred in another aspect of the participant’s life that did not involve their romantic partner. For the pleasant and unpleasant relationship events, many participants reported events such as a happy date or a serious argument, respectively. For the pleasant and unpleasant non-relationship events, many participants reported events such as doing well in a sport or receiving a poor grade on an assignment, respectively.

For each event, participants reported the time of day, the day, and the month that the event occurred, and they wrote a short description of the event, disclosing only as much information as they felt comfortable divulging. This repeated-measures, instructional manipulation occurred within a cross-sectional retrospective study. The participants then reported the way they felt at the time of the event (i.e., their original emotion), the way they felt at the time of the testing (i.e., their current emotion), and the frequency they mentally or verbally rehearsed the event. These factors were reported using the two previously mentioned pleasantness scales and frequency scale, respectively. Unpleasant events were initially rated as negative, and pleasant events were initially rated as positive. The participants completed all questionnaires in approximately 60 min. Finally, the participants were debriefed, and they were asked if they wanted to ask any questions. The questions were answered by researchers. Participants were given the contact information of the head researcher in case they wanted to ask additional questions after the experiment concluded. Participants were also told about the free counseling services on campus provided by the university in the case that they felt a need to utilize those services.

### 3.4. Analytic Strategy

For all analyses, event type was the unit of analysis and fading affect was the dependent variable. We first tested the FAB and whether it was moderated by event type using a 2 (Initial Event Affect) × (Event Type) completely between groups analysis of covariance (ANCOVA) with initial event affect (pleasant or unpleasant) and event type (relationship or non-relationship) as the independent variables and neuroticism and participant as the control variables as carried out in previous research e.g., [22]. We then present results from clustered data, including a nominal-level variable to represent each participant and control for clustered data in each model. This data structure enabled us to test for systematic differences in fading affect among four types of events as they related to self-reported relationship variables (RSES, RPES, RSS, RCS, MAS, FAS, PAS). We tested the two-way interactions between initial event affect (pleasant vs. unpleasant) and individual difference variables in predicting fading affect, while controlling for relevant main effects, and control variables (neuroticism and participant). More importantly, we tested the three-way interaction between initial event affect (pleasant vs. unpleasant), event type (relationship vs. non-relationship), and relationship variables, while controlling for all two-way interactions, relevant main effects, and the control variables.

To test for simple moderation of the FAB, we employed the Process macro via IBM SPSS [51] to examine fading affect, *y*, among pleasant and unpleasant events across the continuum of the relationship variables. Model 1 [51] evaluated the effect of initial event affect, *x*, on fading affect, while controlling for neuroticism and participant, conditional upon levels of self-reported individual difference variables, *w*: RSES, RPES, RSS, RCS, MAS, FAS, and PAS. We used the Johnson-Neyman technique in the Process macro to indicate if the FAB was evident for individuals who reported low or high levels of a particular relationship variable, which avoids drawing an arbitrary line to determine “low” and “high” groups [52].

To test for significant three-way interactions, we again used the Process macro to examine fading affect, *y*, among four categories of events across the continuum of the relationship variables. Specifically, Model 3 [51] enabled the specification of the two-way interaction between initial event affect, *x*, and event type, *m*, while controlling for neuroticism and participant, conditional upon levels of self-reported relationship variables, *w*. We also used the Johnson-Neyman technique to detect specific healthy coping, where the FAB was more strongly related to a relationship variable for one event type (e.g., relationship event) than for another event type (e.g., non-relationship event), indicated by a greater range of effects for the relation of the FAB to a relationship variable for one event type than for the other event type. For any statistically significant finding, at each level of the moderators, we reported the indirect effect, the corresponding standard error, *t*-value, *p*-value, 95% CI lower- and upper-estimates, as well as effect size.

We also evaluated possible mediators of the three-way interaction with the Process macro. Figure 1 illustrates this model. We examined event rehearsal frequency as a mediator of the relation between fading affect and initial event affect, event type, and relationship variables. Process Model 11 [51] enabled the replication of the three-way interaction (i.e., Model 3 is tested within Model 11), as well as examinations of the mediators for this effect. In Model 11, we hypothesized that initial event affect, *x*, predicts differential fading affect, *y*, and the FAB varies across levels of relationship variables, *w*, and event type, *z*, and this effect of *x***w**z on *y* may occur through event rehearsal frequency, denoted as *m* and tested in its own separate model. We reported the conditional indirect effect of *x***w***z* on *y* through *m*, examining the indirect effect of *x* on *y* through *m* at levels of the moderators, *w* and *z*. In each model, we controlled for possible influences due to relationship variables, neuroticism, and participant. We also tested for mediation in any significant three-way interaction.

## 4. Results

### 4.1. Main Effect and Two-Way Interactions: Evidence of General Healthy Coping for FAB

The ANCOVA produced heterogeneity, but violation of this parametric assumption for conducting ANOVA is not a problem if the sample sizes are relatively equal, defined by a ratio of largest to smallest sample sizes equal or less than 1.5 [53]. The sample size ratios calculated for initial event affect, event type, and the interaction were all less than 1.5, and, hence, relatively equal. The analyses controlled for neuroticism and the nominal-level variable for participant; the effect of participant was significant, but the effect of neuroticism was not. The overall analysis of variance was statistically significant, *F*(5, 2706) = 69.336, *p* < 0.001, *η*^2^*_partial_* = 0.114. As expected, an FAB occurred, *F*(1, 2706) = 279.229, *p* < 0.001, *η*^2^*_partial_* = 0.094, such that the affect for unpleasant events (*M* = 1.503, *SE* = 0.044) faded more than the affect for pleasant events (*M* = 0.379, *SE* = 0.044). In addition, affect did not fade differently for relationship events (*M* = 0.911, *SE* = 0.052) than for non-relationship events (*M* = 0.971, *SE* = 0.051), *F*(1, 2706) = 0.476, *p* = 0.490, *η*^2^*_partial_* = 0.000. The two-way interaction between initial event affect and event type was also not statistically significant, *F*(1, 2706) = 0.569, *p* = 0.451, *η*^2^*_partial_* = 0.000. Figure 2 depicts fading affect across initial event affect and event type.

We used the Process Model 1 to examine whether relationship variables predicted the FAB [51]. When examining self-esteem as a predictor, the main effect for initial event affect was not significant, but the main effect of self-esteem was significant. More importantly, the results from the Process Model 1 revealed a significant two-way interaction between initial event affect and self-esteem, B = 0.381 (*SE* = 0.100), *t*(2694) = 3.828, *p* < 0.001, 95% CI [0.186, 0.576], Model *R*^2^ = 0.005, Δ*R*^2^ due to the two-way interaction term, *p* < 0.001. The Johnson-Neyman results suggested that the effect of initial event affect was significant at every level of self-esteem. Figure 3 shows that the FAB increased with self-esteem, mainly because fading affect decreased for pleasant events as self-esteem increased. When examining partner-esteem as a predictor, the main effects of initial event affect and partner-esteem were both significant and the results from the Process Model 1 (Hayes, 2013) revealed a significant two-way interaction between initial event affect (e.g., FAB) and partner-esteem, B = 0.845 (*SE* = 0.112), *t*(2694) = 7.529, *p* < 0.001, 95% CI [0.625, 1.066], Model *R*^2^ = 0.018, Δ*R*^2^ due to the two-way interaction term, *p* < 0.001. The Johnson-Neyman results suggested that the effect of initial event affect was significant when partner-esteem was 1.614 or greater. Figure 4 shows that the FAB increases with partner-esteem, mostly because fading affect decreased for pleasant events as partner-esteem increased.

When examining relationship satisfaction as a predictor, the main effects for both initial event affect and relationship satisfaction were significant, and the results from the Process Model 1 [51] revealed a significant two-way interaction between initial event affect and relationship satisfaction, B = 0.490 (*SE* = 0.061), *t*(2706) = 8.082, *p* < 0.001, 95% CI [0.371, 0.608], Model *R*^2^ = 0.021, Δ*R*^2^ due to the two-way interaction term, *p* < 0.001. The Johnson-Neyman results suggested that the effect of initial event affect became significant when relationship satisfaction was 2.282 and greater. Figure 5 shows that the FAB increased with relationship satisfaction, mostly because fading affect decreased for pleasant events as relationship satisfaction increased. When examining relationship confidence, the main effect of initial event affect and relationship confidence were both significant. More importantly, the results from the Process Model 1 (Hayes, 2013 [51]) revealed a significant two-way interaction between initial event affect and relationship confidence, B = 0.582 (*SE* = 0.119), *t*(2543) = 4.914, *p* < 0.001, 95% CI [0.350, 0.815], Model *R*^2^ = 0.008, Δ*R*^2^ due to the two-way interaction term, *p* < 0.001. The Johnson-Neyman results suggested that the effect of initial event affect became significant when intimate relationship confidence was 2.341 and greater. Figure 6 displays this interaction and shows that the FAB increased with intimate relationship confidence, but only because fading affect for pleasant events decreased with intimate relationship confidence.

When examining mother attachment as a predictor, the main effect for initial event affect was not significant, but the main effect of mother attachment was significant. The results from the Process Model 1 [51] revealed a significant two-way interaction between initial event affect and mother attachment, B = 0.225 (*SE* = 0.075), *t*(2658) = 2.995, *p* < 0.003, 95% CI [0.078, 0.372], Model *R*^2^ = 0.003, Δ*R*^2^ due to the two-way interaction term, *p* < 0.001. The Johnson-Neyman results suggested that the effect of initial event affect was significant at every level of mother attachment. Figure 7 shows that the FAB increased with mother attachment, only because fading affect decreased for pleasant events as mother attachment increased. When examining peer attachment as a predictor, the main effect for initial event affect was not significant, but the main effect of peer attachment was significant. More importantly, the results revealed a significant two-way interaction between initial event affect and peer attachment, B = 0.216 (*SE* = 0.087), *t*(2694) = 2.487, *p* = 0.013, 95% CI [0.046, 0.386], Model *R*^2^ = 0.002, Δ*R*^2^ due to the two-way interaction term, *p* < 0.001. The Johnson-Neyman results suggested that the effect of initial event affect was significant at every level of peer attachment. Figure 8 shows that the FAB increased with peer attachment, because fading affect decreased for pleasant events and it increased for unpleasant events as peer attachment increased.

Table 1 includes the coefficients for the FAB across each of five quintiles for the individual difference variables: self-esteem, partner-esteem, relationship satisfaction, relationship confidence, mother attachment, and peer attachment. As pleasant events were coded as 1 and unpleasant events were coded as 2, large, positive coefficients represent a strong FAB.

### 4.2. Three-Way Interactions: Testing FAB as Healthy Coping at the Specific Level of Analysis

We found significant three-way interactions for partner-esteem, relationship satisfaction, and relationship confidence. For partner-esteem, the effects for the control variables, the main effects, except neuroticism, and the two-way interactions were statistically significant. The Process Model 3 [51] revealed a significant three-way interaction between initial event affect, event type, and partner-esteem, B = −0.774 (*SE* = 0.223), *t*(2690) = −3.464, *p* < 0.001, 95% CI [−1.212, −0.336], Model *R*^2^ = 0.004, Δ*R*^2^ due to the three-way interaction term, *p* < 0.001. Figure 9 shows that the FAB increased significantly with partner-esteem for both relationship and non-relationship events, but this trend was stronger for relationship events, B = 1.241, (*SE* = 0.159), *t*(2690) = 7.812, *p* < 0.001, 95% CI [0.930, 1.553], than for non-relationship events, B = 0.467, (*SE* = 0.157), *t*(2690) = 2.976, *p* = 0.003, 95% CI [0.159, 0.775]. This result showed evidence of specific healthy coping for the FAB. Table 2 includes the coefficients for the FAB per event type across the five quintiles of partner-esteem, with large, positive coefficients representing large FAB. The Johnson-Neyman results indicated that the FAB was significant for both events at every quintile of partner-esteem, except for relationship events at the lowest quintile.

For relationship satisfaction, the effects for the control variables, the main effects, except neuroticism, and the two-way interactions were statistically significant. The Process Model 3 [51] revealed a significant three-way interaction between initial event affect, event type, and relationship satisfaction, B = −0.734 (*SE* = 0.120), *t*(2702) = −6.112, *p* < 0.001, 95% CI [−0.969, −0.498], Model *R*^2^ = 0.152, Δ*R*^2^ due to the three-way interaction term, *p* < 0.001. Figure 10 shows that the FAB increased with relationship satisfaction for both relationship and non-relationship events, but this trend was only significant for relationship events, not for non-relationship events (Figure 10). Table 3 includes the coefficients for the FAB per event type across the five quintiles of partner-esteem, with large, positive coefficients representing large FAB. The Johnson-Neyman results indicated that the FAB was significant for both events at every quintile of relationship satisfaction.

For relationship confidence, the effects for the control variables, the main effects, except for neuroticism, and the two-way interactions were statistically significant. The Process Model 3 [51] revealed a significant three-way interaction between initial event affect, event type, and relationship confidence, B = −0.806 (*SE* = 0.236), *t*(2539) = −3.632, *p* < 0.001, 95% CI [−1.269, −0.344], Model *R*^2^ = 0.133, Δ*R*^2^ due to the three-way interaction term, *p* < 0.001. Figure 11 shows that the FAB increased with relationship confidence for relationship events, but this trend was not significant, and barely noticeable, for non-relationship events (Figure 11). Table 4 includes the coefficients for the FAB per event type across the five quintiles of partner-esteem, with large, positive coefficients representing large FAB. The Johnson-Neyman results indicated that the FAB was significant for both events at every quintile of relationship confidence.

### 4.3. Examining Rehearsal as a Mediator of the Three-Way Interactions

Next, we examined the conditional indirect effects of initial event affect on fading affect for relationship and non-relationship events across quintiles of partner-esteem, relationship satisfaction, and intimate relationship confidence through rehearsal ratings using the Process Model 11 [51]. Table 5, Table 6 and Table 7 show that the relation between the three-way interactions and fading affect was intervened by rehearsal ratings at every quintile of partner-esteem, relationship satisfaction, and intimate relationship confidence across relationship and non-relationship events. However, one notable exception was the lack of mediation for relationship satisfaction at the lowest quintile for relationship events. These results strongly support the model depicted in Figure 1.

## 5. Discussion

### 5.1. FAB and Healthy Coping

We found a robust overall Fading Affect Bias (FAB) that was similar across both relationship and non-relationship events, as demonstrated by the absence of a significant Initial Event Affect by Event Type interaction (*F* < 1). These results displayed general healthy coping for the FAB. Similarly, all but one relationship variable, father attachment, showed general healthy coping for the FAB. Specifically, self-esteem, partner-esteem, relationship satisfaction, relationship confidence, mother attachment, and peer attachment all positively predicted the FAB, which displayed general healthy coping for the FAB. These results replicate most FAB research produced in the literature showing general healthy coping [17,19,20,21,40,42]. The FAB is a form of general healthy coping because it is properly connected to healthy/unhealthy variables. Moreover, the FAB is a form of emotion regulation because it makes people feel good about their experiences, which enhances their perceptions of themselves [15], supporting self-enhancement theories (e.g., [26]).

As expected, most of the continuous relationship variables positively predicted the FAB. The positive relations of the FAB to both self-esteem and partner-esteem were expected because self-esteem is positively related to healthy outcomes [21,54,55], including the FAB [22], and partner-esteem, which was based on self-esteem, is also positively related to healthy outcome variables [56,57,58]. Similarly, the positive relations of the FAB to both relationship satisfaction and relationship confidence were expected because both relationship measures are positively related to healthy outcome variables [31,32,55]. In addition, secure attachments were expected to positively predict the FAB because secure attachments are related to healthy outcome variables [34,35,36,59].

Both mother attachment and peer attachment positively predicted the FAB, but father attachment was not related to the FAB. The significant relations between mother and peer attachment and FAB indicate that secure connections to one’s mother and friends are important to emotionally regulate via the FAB and feel good about one’s self. In contrast, the attachment with one’s father apparently does not seem to help individuals regulate their emotions in the same way, which suggests that fathers do not interact with their children in ways that foster emotion regulation, at least not in the form of FAB. As Muir and colleagues [18] found that social rehearsals increase the FAB when the listener was active rather than passive, one possible explanation for the lack of a relation between father esteem and the FAB may be that fathers are passive as opposed to active listeners. Future research could test this possibility by replicating the current study and testing the degree that mothers, fathers, and peers are active listeners. If active listening explains the different relations between attachment and the FAB, mothers and peers should be rated higher on active listening than fathers and active listening should mediate the relations between mother and peer attachment and the FAB.

### 5.2. FAB Shows First Instance of Specific Healthy Coping

In addition to the two-way interactions, three three-way interactions were significant; partner-esteem, relationship satisfaction, and relationship confidence each interacted with initial event affect and event type to predict fading affect. The FAB was positively related to partner-esteem for both relationship and non-relationship events, but the effect was stronger for relationship events. As the FAB showed differential healthy coping across types of events, this result demonstrated the first instance of specific healthy coping in the FAB literature. As described in the introduction, specific healthy coping is rare because healthy coping is still shown for both events, but it is larger for one event than the other event. In contrast, the three-way interactions in the FAB literature prior to this finding showed healthy coping for one event and no healthy coping or unhealthy coping for the other event. As examples of this common absence of specific healthy coping in the current study, the FAB was positively related to both relationship satisfaction and relationship confidence for relationship events, but not for non-relationship events. The absence of specific healthy coping for the FAB has also been the case in past research for alcohol events [42], religious events [27], communal/achievement events [25], death events [41], social media events [22], and relationship events [3].

### 5.3. Potential Importance of Partner-Esteem and Related Variables

The significant three-way interaction with partner-esteem that indicates specific healthy coping makes one wonder why partner-esteem is special enough to significantly, appropriately (positively), and differentially predict the FAB for relationship and non-relationship events when no other variable has done so in the current study or the literature. One factor that may help explain the predictive power of partner-esteem in terms of the FAB could be trust. Luchies and colleagues [5] showed a relation between high trust in one’s partner and willingness to forgive the partner and, more importantly for the FAB, failure to recall the severity of their transgressions. In other words, trust could have led to strong fading of unpleasant affect for relationship events, which means that trust may be a critical component of partner-esteem, especially for relationship events. Future research should replicate the procedures used in the current study and add partner-trust to test if it helps explain (i.e., mediate) the relation of partner-esteem and the FAB.

As partner-esteem is beneficial in regulating emotions tied to relationship and non-relationship events, future research should also test interventions (e.g., therapeutic techniques) to enhance partner-esteem and test its effect on FAB and emotional measures (e.g., anxiety, depression) across relationship and non-relationship events. In fact, the relation should be tested across a variety of events and their control events to determine if increases in partner-esteem can increase the FAB across events that do not involve relationships. Partner-esteem may be special enough to enhance emotion regulation in the form of the FAB across events that are completely distinct from relationships, such as communal events, religious events, sporting events, and alcohol events to name just a few events. If partner-esteem consistently and positively predicts the FAB across a variety of events and contexts, the phrase “happy wife, happy life” could be replaced by “partner-esteem, living the dream”.

### 5.4. Rehearsals Mediate Three-Way Interactions for the FAB

The FAB was mediated by event rehearsals at every quintile of partner-esteem, relationship satisfaction, and relationship confidence across relationship and non-relationship events with one exception: event rehearsals did not mediate relationship satisfaction at the lowest level for relationship events. Combined with past findings, these results make the point that rehearsal is important for the FAB [16,17] and it is a consistent explanatory mechanism across several FAB studies e.g., [22,42]. In the current study, partner-esteem, relationship satisfaction, and relationship confidence likely increased rehearsal of events, which increased the fading of affect for unpleasant events and inhibited the fading of affect for pleasant events with the rehearsal effect being larger for relationship events than for non-relationship events. Future research could test this explanation empirically by replicating the current experiment and manipulating event rehearsals. Specifically, participants could be asked to rehearse some of their events frequently. If event rehearsal is an explanatory factor for FAB effects, as suggested by the mediation results of the current study and past studies, the relation of FAB and relationship variables (e.g., partner-esteem, relationship satisfaction, and relationship confidence) should strengthen for both relationship and non-relationship events when events are deliberately rehearsed.

The only overlap between the current study and the relationships and FAB study conducted by Zengel and colleagues [3] is that they both evaluated FAB for relationship events and they both examined attachment. Both studies showed significant FAB effects that persisted across event types and both studies showed relations of the FAB to attachment. In terms of attachment, Zengel and colleagues separated the measure into five categories, whereas we simply wanted to measure attachment as a continuous measure ranging from insecure to secure. Our goal was to examine and attempt to display specific healthy coping for the FAB in the context of relationship events, and continuous measures facilitated this goal. Therefore, we used continuous attachments to participants’ mother, father, and a peer ranging from insecure to secure. Categorizing participants into separate, small groups would not have aided our goals. Although we did not find specific healthy coping for FAB with any of the attachment measures, the continuous measures gave us the best opportunity possible to show that outcome, and both mother and peer attachment successfully predicted strong FAB.

Data were lost in the current study because participants did not always follow directions, even though we took many steps, based on previous research, to ensure that they would follow instructions while maintaining their confidentiality. On a positive note, the loss of data was minimal, and the sample size was large because we gathered three events from each category created by crossing initial event affect and event type. Although we usually ask participants to provide two events from each category, we wanted the extra power to detect specific healthy coping in the form of three-way interactions. As mentioned previously, we wanted to provide favorable circumstances to allow and detect specific healthy coping for the FAB, and the effort was justified, because the outcome was successful. Partner-esteem was the first variable in the FAB literature to predict the FAB in the appropriate way (positive for healthy variables and negative for unhealthy variables) and show healthy coping for both event types with this relation being stronger for one event than for the other event. Specifically, partner-esteem positively predicted the FAB for both relationship and non-relationship events, but the relation was stronger for relationship events than for non-relationship events.

## 6. Conclusions

The FAB was equally robust for both relationship and non-relationship events, which showed general healthy coping. In addition, several healthy relationship variables positively predicted the FAB, which demonstrated general healthy coping and replicated past research. The general healthy coping tied to the FAB regulates emotions and it enhances perceptions of the self. In addition, peer-esteem, relationship satisfaction, and relationship confidence all positively predicted the FAB for relationship events, but only partner-esteem positively predicted the FAB across non-relationship events, making partner-esteem the first variable to show specific healthy coping for the FAB in the literature. Of similar importance, event rehearsals mediated all three of the three-way interactions and they did so at every quintile, except for one, which means that event rehearsals continue to explain complex FAB effects. Future research should explore methods to deliberately increase partner-esteem and event rehearsals to increase emotion-regulating, self-enhancing FAB, as these techniques could help people identify as someone who values their partner and perceives their events as worthy and central enough to the self to think about and share. In closing, partner-esteem and event rehearsals seem to be important variables for emotion regulation in the form of the FAB, and future research should determine other similarly healthful variables.

## Figures and Tables

**Figure 1 ijerph-18-10121-f001:**
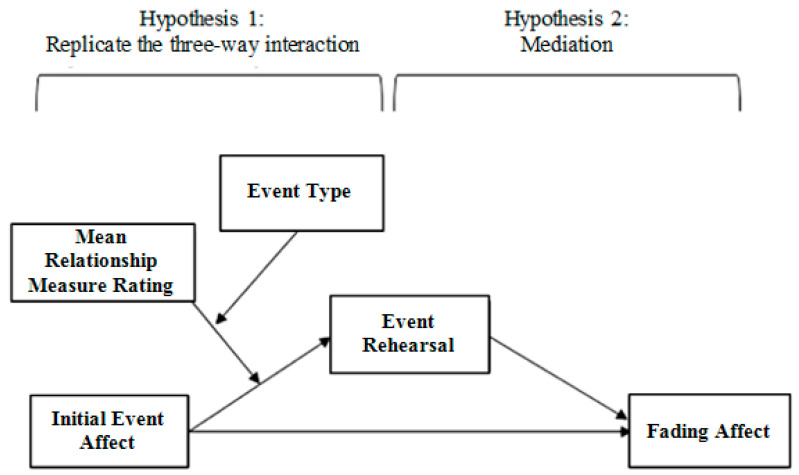
The three-way interaction between initial event affect, event type, and a relationship variable (e.g., relationship satisfaction) predicts fading affect through the mediatior of event rehearsal frequency.

**Figure 2 ijerph-18-10121-f002:**
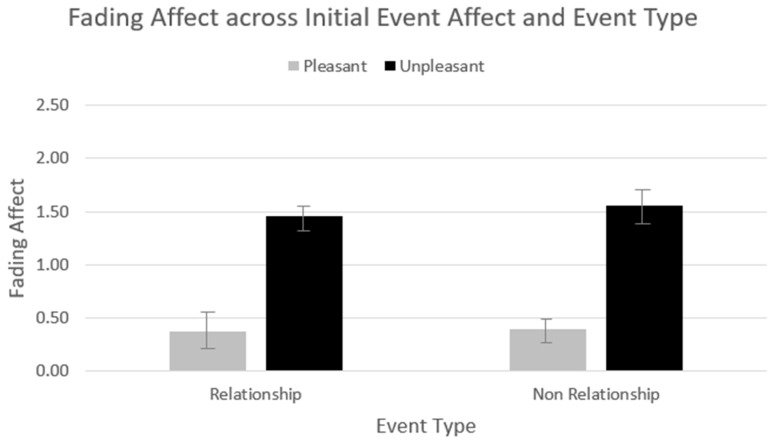
Fading affect across pleasant and unpleasant relationship and non-relationship events.

**Figure 3 ijerph-18-10121-f003:**
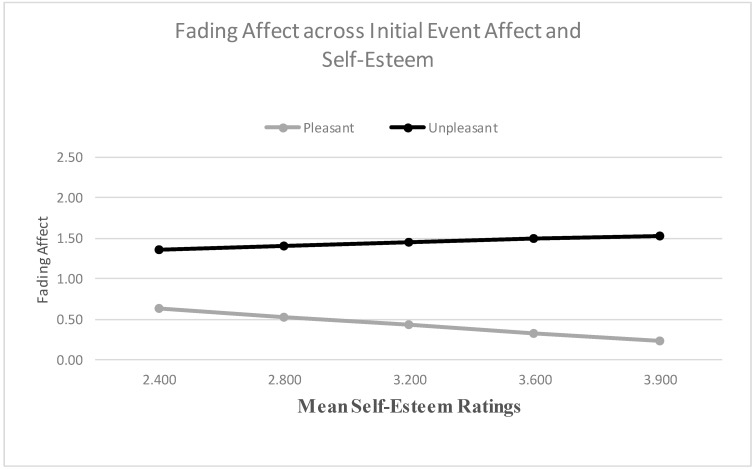
Fading affect for pleasant and unpleasant events across quintile levels (10th through 90th) of self-esteem.

**Figure 4 ijerph-18-10121-f004:**
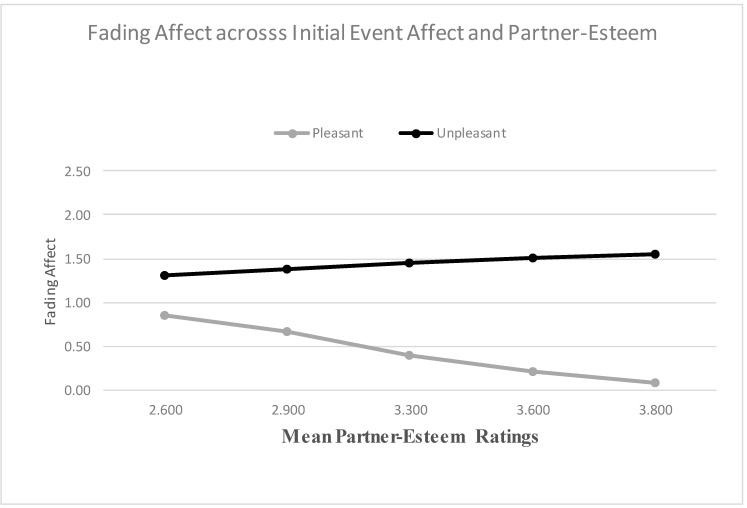
Fading affect for pleasant and unpleasant events across quintile levels (10th through 90th) of partner-esteem.

**Figure 5 ijerph-18-10121-f005:**
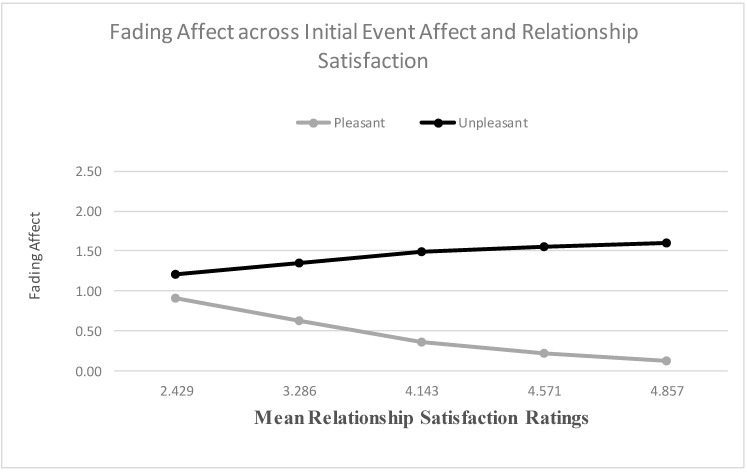
Fading affect for pleasant and unpleasant events across quintile levels (10th through 90th) of relationship satisfaction.

**Figure 6 ijerph-18-10121-f006:**
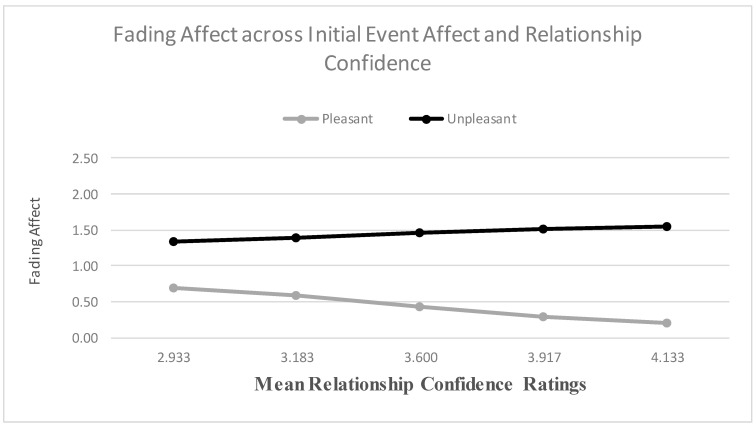
Fading affect for pleasant and unpleasant events across quintile levels (10th through 90th) of relationship confidence.

**Figure 7 ijerph-18-10121-f007:**
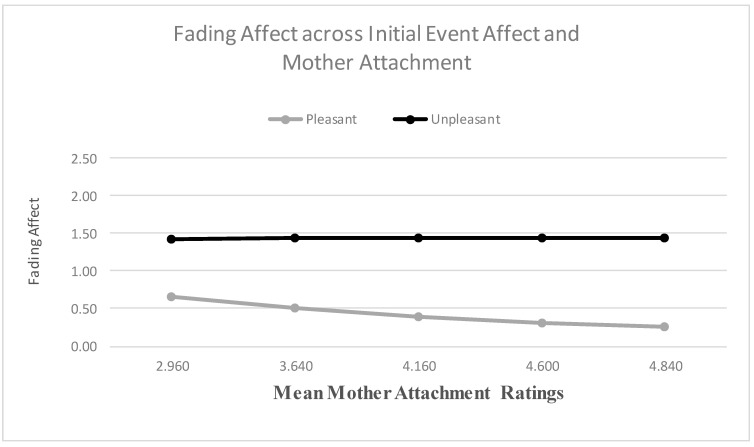
Fading affect for pleasant and unpleasant events across quintile levels (10th through 90th) of mother attachment.

**Figure 8 ijerph-18-10121-f008:**
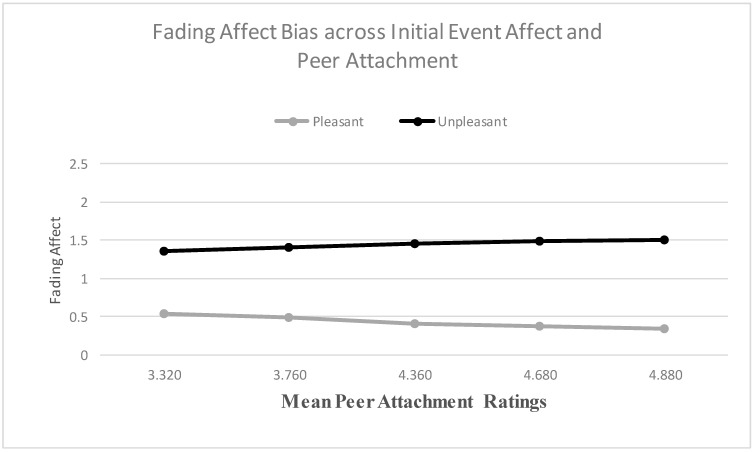
Fading affect for pleasant and unpleasant events across quintile levels (10th through 90th) of peer attachment.

**Figure 9 ijerph-18-10121-f009:**
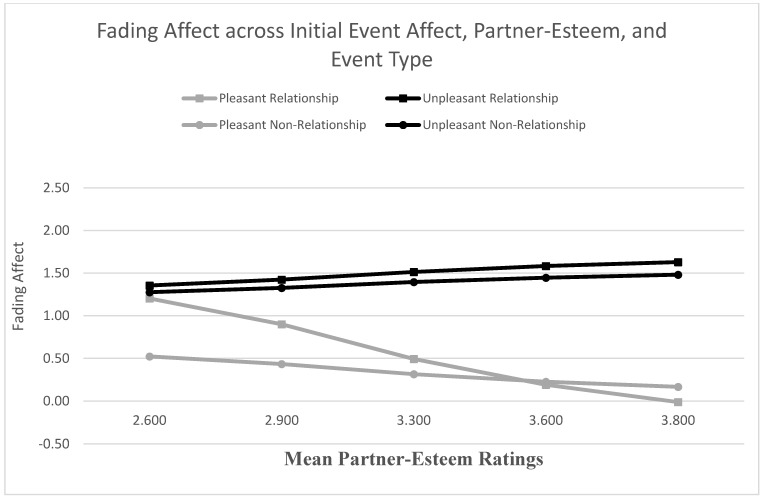
Fading affect for pleasant and unpleasant relationship and non-relationship events across quintile levels (10th through 90th) of partner-esteem.

**Figure 10 ijerph-18-10121-f010:**
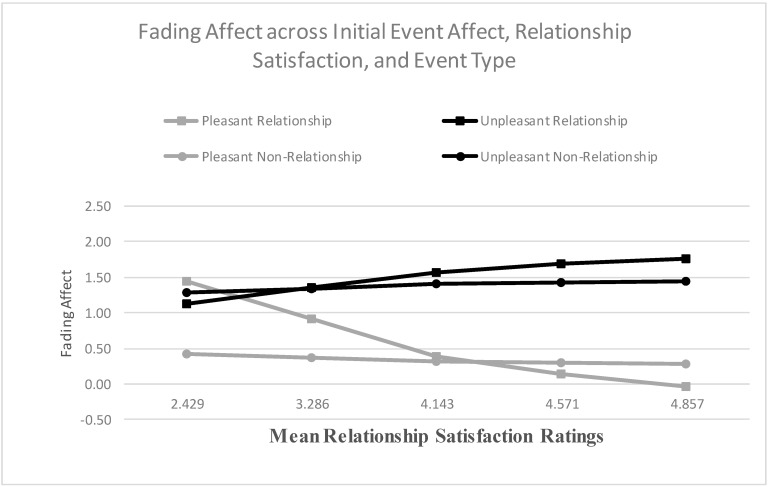
Fading affect for pleasant and unpleasant relationship and non-relationship events across quintile levels (10th through 90th) of relationship satisfaction.

**Figure 11 ijerph-18-10121-f011:**
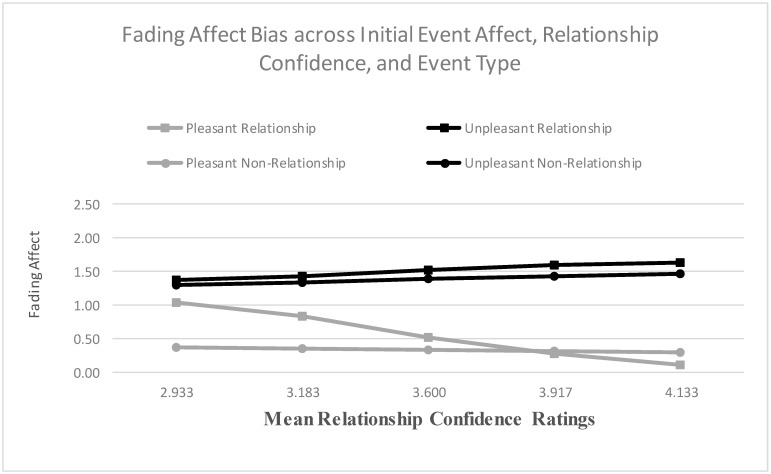
Fading affect for pleasant and unpleasant relationship and non-relationship events across quintile levels (10th through 90th) of relationship confidence.

**Table 1 ijerph-18-10121-t001:** Fading Affect Bias (FAB) Regression Coefficients (SE) across Quintiles of Self-Esteem, Partner-Esteem, Relationship Satisfaction, Relationship Confidence, Mother Attachment, and Peer Attachment Ratings (large positive coefficients represent large FAB).

Measures	Quintile Regression Coefficients Ranging from 10 to 90				
	10th	25th	50th	75th	90th
Self-Esteem MR RC	2.400 0.719 (0.092)	2.800 0.871 (0.065)	3.200 1.024 (0.056)	3.600 1.176 (0.072)	3.900 1.290 (0.094)
Partner-Esteem MR RC	2.600 0.460 (0.090)	2.900 0.714 (0.067)	3.300 1.052 (0.063)	3.600 1.305 (0.068)	3.800 1.474 (0.084)
Relationship Satisfaction MR RC	2.429 0.289 (0.104)	3.286 0.709 (0.065)	4.143 1.129 (0.057)	4.571 1.339 (0.069)	4.857 1.478 (0.081)
Relationship Confidence MR RC	2.933 0.648 (0.093)	3.183 0.794 (0.072)	3.600 1.037 (0.057)	3.917 1.221 (0.071)	4.133 1.347 (0.089)
Mother Attachment MR RC	2.160 0.844 (0.103)	2.880 0.916 (0.070)	3.720 0.999 (0.057)	4.320 1.058 (0.072)	4.720 1.098 (0.089)
Peer Attachment MR RC	3.320 0.822 (0.093)	3.760 0.917 (0.066)	4.360 1.046 (0.058)	4.680 1.116 (0.071)	4.880 1.159 (0.083)

Notes. MR = Mean Ratings and RC = Regression Coefficients (SE). All *ps* < 0.005.

**Table 2 ijerph-18-10121-t002:** Fading Affect Bias Regression Coefficients (SE) for Relationship Events and Non-Relationship Events across Quintiles (Mean Rating) of Partner-Esteem Ratings (large positive coefficients represent large FAB).

Quintile (Mean Partner-Esteem Ratings)	Event Type	
	Relationship Events	Non-Relationship Events
10th (2.600)	0.151 (0.151) +	0.753 (0.126)
25th (2.900)	0.523 (0.095)	0.893 (0.094)
50th (3.300)	1.020 (0.078)	1.080 (0.078)
75th (3.600)	1.392 (0.097)	1.220 (0.096)
90th (3.800)	1.640 (0.118)	1.313 (0.117)

Notes. + *p* > 0.05. All other *ps* less than 0.001.

**Table 3 ijerph-18-10121-t003:** Fading Affect Bias Regression Coefficients (SE) for Relationship Events and Non-Relationship Events across Quintiles (Mean Rating) of Relationship Satisfaction Ratings (large positive coefficients represent large FAB).

Quintile (Mean Relationship Satisfaction Ratings)	Event Type	
	Relationship Events	Non-Relationship Events
10th (2.429)	−0.311 (0.147) *	0.850 (0.143)
25th (3.286)	0.432 (0.093)	0.965 (0.091)
50th (4.143)	1.175 (0.080)	1.079 (0.080)
75th (4.571)	1.547 (0.097)	1.136 (0.096)
90th (4.857)	1.794 (0.114)	1.174 (0.113)

Notes. * *p* < 0.05. All other *ps* less than 0.001.

**Table 4 ijerph-18-10121-t004:** Fading Affect Bias Regression Coefficients (SE) for Relationship Events and Non-Relationship Events across Quintiles (Mean Rating) of Relationship Confidence Ratings (large positive coefficients represent large FAB).

Quintile (Mean Relationship Confidence Ratings)	Event Type	
	Relationship Events	Non-Relationship Events
10th (2.933)	0.336 (0.133) *	0.942 (0.130)
25th (3.183)	0.585 (0.102)	0.989 (0.101)
50th (3.600)	0.999 (0.081)	1.068 (0.080)
75th (3.917)	1.314 (0.100)	1.128 (0.100)
90th (4.133)	1.530 (0.126)	1.169 (0.125)

Notes. * *p* < 0.05. All other *ps* less than 0.001.

**Table 5 ijerph-18-10121-t005:** Conditional Indirect Effects of Fading Affect Bias (FAB) for Relationship and Non-Relationship Events across Quintiles of Self-Reported Partner-Esteem Ratings through Event Rehearsal Frequency Ratings.

Quintile (Mean Partner-Esteem Ratings)	Event Type	
	Relationship Events	Non-Relationship Events
10th (2.600)	0.034 (0.013)	0.068 (0.020)
25th (2.900)	0.053 (0.015)	0.072 (0.019)
50th (3.300)	0.079 (0.021)	0.076 (0.020)
75th (3.700)	0.105 (0.029)	0.080 (0.022)
90th (3.800)	0.111 (0.031)	0.082 (0.023)

Notes. All *ps* less than 0.05.

**Table 6 ijerph-18-10121-t006:** Conditional Indirect Effects of Fading Affect Bias (FAB) for Relationship and Non-Relationship Events across Quintiles of Self-Reported Relationship Satisfaction Ratings through Event Rehearsal Frequency Ratings.

Quintile (Mean Relationship Satisfaction Ratings)	Event Type	
	Relationship Events	Non-Relationship Events
10th (0.000)	0.011 (0.013) +	0.066 (0.021)
25th (0.143)	0.048 (0.014)	0.071 (0.020)
50th (0.429)	0.086 (0.024)	0.077 (0.021)
75th (0.857)	0.105 (0.029)	0.079 (0.022)
90th (1.286)	0.117 (0.033)	0.081 (0.023)

Notes. + *p* > 0.05. All other *ps* less than 0.05.

**Table 7 ijerph-18-10121-t007:** Conditional Indirect Effects of Fading Affect Bias (FAB) for Relationship and Non-Relationship Events across Quintiles of Self-Reported Relationship Confidence Ratings through Event Rehearsal Frequency Ratings.

Quintile (Mean Relationship Confidence Ratings)	Event Type	
	Relationship Events	Non-Relationship Events
10th (2.933)	0.045 (0.015)	0.080 (0.022)
25th (3.183)	0.060 (0.017)	0.081 (0.021)
50th (3.600)	0.086 (0.022)	0.083 (0.021)
75th (3.917)	0.105 (0.027)	0.085 (0.022)
90th (4.133)	0.118 (0.031)	0.087 (0.023)

Notes. All *ps* less than 0.05.

## Data Availability

The data that support the findings of this study are available from the corresponding author upon reasonable request.

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
