# Peer review of "A Fading Affect Bias First: Specific Healthy Coping with Partner-Esteem for Romantic Relationship and Non-Relationship Events"

_ijerph, 2021, doi:10.3390/ijerph181910121_

Round 1

Reviewer 1 Report

it is a good work, well planned and developed

Here are some suggestions:

no connection is found between introduction and discussion, which would be interesting

the statistical analysis section is excessively long, given the objectives of the work, some key questions can be analyzed that would help the reader to understand the manuscript

APA regulations have not been followed

kind regards

Author Response

Reviewer 1 said “it is good work, well planned and developed”.  We appreciate the positive comment.

Reviewer 1 said “no connection is found between the introduction and discussion, which would be interesting”.  We adamantly disagree with the first part of this statement because the introduction describes the FAB with past research, we talk about the fact that the FAB indicates healthy coping because it is positively related to healthy outcomes and negatively related to unhealthy outcomes.  We then talk about the fact that the FAB has consistently shown general healthy coping, but it has not shown specific healthy coping.  In fact, we talk about the fact that a recent study on relationships and FAB has not shown specific healthy coping, possibly because the study did not include non-relationship events.  Therefore, we state that we are conducting the current study examining the relation of relationship variables to the FAB across relationship and non-relationship events to show evidence of specific healthy coping.  In the discussion, we describe the fact that we found the FAB, we show that the FAB was positively related to several relationship variables, which shows general healthy coping.  We then describe several three-way interactions.  Although two of the three-way interactions show general healthy coping, the one involving partner-esteem shows specific healthy coping, which is the first time such a finding has occurred in FAB research.  These facts should make the point that the introduction and the discussion are intricately tied together.  For a specific example of this connection, lines 93-95 in the Introduction asserts that FAB research has not yet shown healthy coping at the specific level of analysis, which is then tied to lines 685-691 of the Discussion where we review this studies’ results of specific healthy coping.

Reviewer 1 said that “the statistical analysis section is excessively long”.  We agree that it is long, but it is long because it is thorough.  Prior to submission, we trimmed the results and made them as lean as possible, and we double checked them for places to cut after this review and all the content is relevant.  Removing any of the text in the Results section would extract critical pieces of information that need to be presented, thereby diminishing their veracity and the quality of the presentation. Importantly, the results along with the figures and tables look excellent in the format used for the paper by IJERPH’s editing team, and they look less long than in MS Word. 

Reviewer 1 said that “some key questions can be analyzed that would help the reader understand the manuscript”.  The main questions were: Is an FAB present?; Does the FAB show healthy coping with relationship variables?; Does the FAB show specific healthy coping with relationship variables across relationship and non-relationship events?  The results answer these questions.  Does Reviewer 1 believe that we should eliminate some of the analyses that answer these questions?  If Reviewer 1 would point out the questions and analyses that they perceive as irrelevant and convince us that those questions and analyses are unimportant, we would eliminate them.  However, we strongly believe that every analysis included in the study targets and answers an important research question.

Reviewer 1 says that “APA regulations have not been followed”.  As the first author teaches APA format in a Research Methods in Psychology class and knows it very well, we did not understand where such errors could have occurred, but we think we found the errors in the references.  The formatting errors occurred because the first author tabbed three references to align them, and then the manuscript processing misaligned those three references.  We have now corrected those references. 

Reviewer 2 Report

This manuscript furthers past work on the fading affect bias, by including stronger tests of its relationship to healthy coping behaviors. It is generally well-written, and advances work in this area.

There are a few minor issues that should be addressed, outlined below.

On line 245, the authors state "We controlled for neuroticism and participant because..." but a word appears to be missing after participant.

On line 300, a reference for the Rosenberg Satisfaction Scale is needed.

On line 306, a reference for the Relationship Confidence Scale is needed.

On lines 347-351, the authors refer to a decision to "analyze the data in all the different ways that affect can change or remain stable"; however, whenever a decision of this sort is made (based on assumptions about how the analysis might be affected), it is better to be sure. Thus, the authors should present a synopsis of any major changes seen by different analytic strategies (if of little significance, in a footnote).

Author Response

Reviewer 2 said “This manuscript furthers past work on the fading affect bias, by including stronger tests of its relationship to healthy coping behaviors. It is generally well-written, and advances work in this area.”  We agree with Reviewer 2, and we would like to thank him/her for their acknowledgment of that fact.

Reviewer 2 says that a word is missing after participant.  No word is missing, and the prose works well as it is stated, but we put in “the nominal participant variable” for “participant” to enhance clarity.

Reviewer 2 points out that we missed 2 references for 2 scales.  We thank Reviewer 2 for catching these omissions and we have rectified these mistakes.

Reviewer 2 talks about the fact that we talk about analyzing either all the data or part of the data.  The fading affect measure could only include affect that does or does not fade or it could include all the possible changes, including increases in affect and affect that changes from pleasant to unpleasant or affect that changes from unpleasant to pleasant.  This point is a philosophical issue that affects the results.  The FAB research in general and the FAB research examined by Gibbons and colleagues has typically analyzed all the data, which involves all the ways that affect can change, not just part of the data that conforms to a restrictive definition of fading.  We reference Skowronski who suggested that it is perfectly acceptable to use the measure we used.  Moreover, the choice to analyze all the ways that affect can change limits the bias from a narrow definition of fading affect and it leads to high statistical power and high generalizability, whereas the narrow definition of fading affect leads to low statistical power and, more importantly, reduced generalizability.  In other words, we analyzed the data with a measure that makes the most sense and the findings from this measure are more meaningful and important than the findings from a more restrictive measure.  We addressed this issue briefly in the fading affect and rehearsal for events section of the paper.   

Reviewer 3 Report

Dear author

This manuscript aims to understand the impact of emotions between men and women on people’s health. This will help promote people's interaction and health in life, which is an interesting topic.

The overall suggestion is: to overthrow the theory and the order of the literature, and re-plan. The overthrow of research theories, explanations and research frameworks need to be reconfirmed. Please exclude literature in conclusion. Compile the layout according to the journal's researcher submission guidelines.

  1. About the introduction and literature review

Although the introduction is very concise, but an easy-to-read manuscript, the content presented should be related to the topic and have reasonable logic.

a). Is it necessary to present the descriptions of crime, homosexuality, alcoholism, etc. in the manuscript? It is recommended to explain appropriately or delete relevant words (because these are not themes).

b). In the description of the literature reviewed in the manuscript, it is not easy to understand that the theory or evidence presented in the reviewed literature is related to the topic. This part recommends whether the author combines the theory in the text with the actual research results, so that it is easier for readers to read.

c). The final description of this paragraph is inconsistent with the research framework diagram, which is confusing.

  1. There are about methods

Usually the research process and structure will be explained in the initial stage of the method. Or, in the method chapter, a clear research architecture diagram appears from the beginning. Obviously, the author's approach is different, and because of this, it is not easy to understand the researcher's theme and main discussion direction.

a). It is recommended to adjust the order.

b). In the number, please unify the number presentation method for the numbers related to ".002" and the subsequent "0.002".

  1. About Discussion

It is recommended to include sub-headings based on the topics of the analysis.

  1. About Conclusions

Usually the conclusion only presents the findings or contributions of the research, it is recommended not to include the text in this chapter now. Then, based on the findings of the manuscript, what research recommendations are there in the future?

I hope that the author can respond or adjust according to the suggestions, which will help the article to improve the visualization.

Wish you all the best

Author Response

Reviewer 3 said “The manuscript aims to understand the impact of emotions between men and women on people’s health. This will help promote people’s interaction and health in life, which is an interesting topic:”  We agree and thank Reviewer 3 for their insight and positive comment.

Reviewer 3 said that we should “compile the layout according the journal’s submission guidelines.”  The Instructions for Authors on the IJERPH website said to provide a Title, Authors and Affiliations, Abstract, Keywords, followed by an Introduction, Method, Results, Discussion, and Conclusion, followed by Funding, an Institutional Review Board Statement, an Informed Consent Statement, a Data Availability Statement, Acknowledgements, Conflicts of Interest, and References.  We followed that exact layout.    

Reviewer 3 said “the overall suggestion is: to overthrow the theory and the order of the literature, and re-plan.  The overthrow of research theories, explanations and research frameworks need to be reconfirmed.”  Reviewer 3 later said “Although the introduction is very concise, but an easy-to-read manuscript, the content presented should be related to the topic and have reasonable logic.” Reviewer 3 also said “In the description of the literature reviewed in the manuscript, it is not easy to understand that the theory or evidence presented in the reviewed literature is related to the topic.  This part recommends whether the author combines the theory in the text with the actual research results, so that it is easier for readers to read.” 

In short, Reviewer 3 is saying that the introduction is not set up logically and that the research presented is not relevant to the paper.  We adamantly disagree.  We want Reviewer 3 to know that we put in great effort to properly organize the first paragraph and the rest of the introduction.  We used the first paragraph to mention relationships, grab the readers’ attention, create a roadmap for the rest of the introduction, and state the goal.  After the goal statement in the first paragraph, we talk about FAB research and the fact that the FAB is positively related to pleasant/healthy outcomes and it is negatively related to unpleasant/unhealthy outcomes, which is healthy coping.  We then mention the fact that most FAB studies have shown general healthy coping where the FAB is significant for experimental events and non-significant or reversed for control events, but they have not shown specific healthy coping where the FAB is significant for both events but it is also larger for experimental than control events.  We then mention the fact that one past study examined the FAB in the context of relationships, but it did not show specific healthy coping because it did not use non-relationship events.  As relationships were introduced, we described research examining relationship variables that are positively related to pleasant/healthy variables and negatively related to unpleasant/unhealthy, because these variables would be most likely to predict the FAB.  When describing the relationship variables, we started with self-esteem because a person’s self-worth greatly affects their relationships, and then we describe partner-esteem, because relationships demand partners and a person’s view of their partner determines the way they will act towards and feel about their partner.  We then describe relationship satisfaction, which is an important variable for relationships.  Relationship satisfaction is related to relationship confidence and attachment, which followed relationship satisfaction.  We believe that these sections are logically ordered, but we are open to suggestions for a better order if the reviewers can provide such an order and a rationale for it.  After describing the relationship variables, we summarized the research briefly, and we described the study and the hypotheses, which is also a logical way to set up an experiment.   

Reviewer 3 says “Is it necessary to present the descriptions of crime, homosexuality, alcoholism, etc. in the manuscript?  It is recommended to explain appropriately or delete relevant words (because these are not themes).”  We are not sure why Reviewer 3 mentioned crime.  We agree that it does not need to be mentioned in the paper.  We mention homosexuality because one author examined gay, lesbian, and married heterosexual individuals, who provided the same results, and that research was relevant for the research on partner-esteem.  We mention alcohol for one article that examined the FAB across alcohol consumption as well as alcohol and non-alcohol events.  This study was the seminal study evaluating FAB across individual difference variables (religion, social media, videogame addiction) and related event types.  Therefore, the inclusion of this research was very important.  

Reviewer 3 said “The final description of this paragraph is inconsistent with the research framework diagram, which is confusing.”  We are sorry that Reviewer 3 is confused but the Figure shows that initial affect (pleasant/unpleasant), event type (relationship/non-relationship), and relationship variables (e.g., partner-esteem or relationship satisfaction) combine to influence fading affect but rehearsal ratings intervene and mediate the relation of the three predictors and fading affect.  This Figure perfectly depicts the most important predictions in the experiment, which we accurately described in that paragraph. 

Reviewer 3 talks about methods including a diagram to describe the methods overall and Reviewer 3 says the order described in the Method is incorrect.  We followed APA format. As APA format directs researchers to describe Participants, Materials, and Procedure in papers in that order, we include those sections in that order.  However, Reviewer 3’s comment prompted us to add a brief overall description of the questionnaires at the beginning of the Materials, which also describes the order that participants complete those questionnaires in the Procedure.  We think that this brief description takes the place of the diagram that Reviewer 3 suggested and it provides the reader with a nice overview of the Materials and Procedure.

Reviewer 3 said that we should present leading 0s in front of p-values, but the APA Manual says that numbers that cannot exceed 1 or -1 should not use leading 0s.  We checked the Instructions for Authors on the IJERPH website and it did not say to present leading 0s.  If we missed this instruction somewhere else on the website, we can certainly make those changes. 

Reviewer 3 directed us to add sub-headings in the Discussion and we added them.  Reviewer 3 said to “exclude literature in conclusion”, so we deleted references in the conclusion.

Reviewer 3 said “Usually the conclusion only presents the findings or contributions of the research, it is recommended not to include the text in this chapter now. Then, based on the findings of the manuscript, what research recommendations are there in the future?”  We are very confused about these potentially conflicting statements about not including text that is normally included.  Every paper should conclude with a paragraph that summarizes the findings and implications and then end the paper with a closing sentence that leaves the reader thinking about the topic, which we did in the Conclusion section, which is mandatory for IJERPH.      

Round 2

Reviewer 3 Report

Dear author
It is great to see the adjustment of the manuscript.
Perhaps, there may be different opinions in terms of conveying expressions and research experience, but I believe that the current manuscript is clearer.

I think it has reached the level of publication.

Congratulations 

Author Response

Thanks for your help.